# Grazing Management, Forage Production and Soil Carbon Dynamics

**Mark E. Ritchie**

Department of Biology, Syracuse University, Syracuse, NY 13244, USA; meritchi@syr.edu; Tel.: +315-447-1612; Fax: +315-443-2012

**Abstract:** Soil carbon pools remain a target for sequestering greenhouse gases, but appropriate land management options to achieve such sequestration remain uncertain. Livestock grazing can have profound positive or negative effects on soil carbon. Different models for assessing the influences of grazing are available, but few explicitly account for different management options on soil organic carbon (SOC). Here, I link a previous simple SOC dynamic model (SNAP) to a recent model of episodic grazing and its effects on primary production. The resulting combined model, called SNAPGRAZE, assesses the potential effects of grazing management on SOC across a range of climates with only eight climate, soil, and management input variables. SNAPGRAZE predicts that, at high stocking densities relative to those sustainable under continuous grazing and at higher mean annual temperature and precipitation, short-duration, high stocking density (SDHSD) grazing schemes can enhance forage production and increase stocks of soil organic carbon. Model predictions for current SOC, given a known 50 year grazing history, agrees well with data from nine private ranches in the North American Great Plains. SNAPGRAZE may provide a framework for exploring the consequences of grazing management for forage production and soil carbon dynamics.

**Keywords:** soil; carbon; models; grazing; grasslands; temperate; rotational grazing; livestock

## 1. Introduction

Climate change induced partially by accumulation of greenhouse gases in the atmosphere represents a critical threat to ecosystems and human livelihoods [1–3]. Sequestering $CO_2$, the most abundant greenhouse gas, from the atmosphere to stable pools, such as soils, sediments and wood, represents a significant opportunity to mitigate climate change [4–6]. Soils are increasingly recognized as a potential pool in which to sequester $CO_2$, but considerable debate persists as to appropriate land management actions that can achieve significant sequestration at scale [4,7,8].

Grazing by livestock is perhaps the most widespread use of grassland ecosystems worldwide [9–12] and is fundamental to human livelihoods. Consequently, a key question is how or whether livestock can be managed to sequester carbon and still support human livelihoods. Even if appropriate management strategies can be devised, there is still uncertainty as to how much carbon is gained or lost from their implementation. To that end, appropriate and tested soil carbon dynamic models that incorporate grazing effects on production, biomass, nutrient cycling and other well-known interactions [5,6,13] are needed to confidently assess effects of different management actions.

To achieve this broad goal, soil carbon dynamics need to be linked to realistic descriptions of different management actions. Grazing management occurs as decisions by livestock holders about the nature of grazing "episodes"—that is, the number of animals, of what size, that graze for some time length in an area (pasture). Older models that consider animal numbers multiplied by time, or animal unit months (AUMs), for a given pasture offer limited insight into more modern approaches that track stocking density, frequency and duration of grazing episodes, or "periods of stay", in "rest-rotation",

"mob", "cell", or other similarly labelled grazing systems [14–16]. Ecological models of herbivory and grazing have not informed this diversity of management approaches [17–19]. One model, called GRASIM [20–22], explores influences of daily updated grazing on forage production and C and N cycling in intensively managed, fertilized pastures, but does not explicitly explore the effects of grazing periods of stay and may have limited application to a broad range of grassland types.

　　　Here, I build on a recent episodic herbivory model, called EHM [23], to generate a framework for grazing management decisions and their effects on grassland production and soil carbon. Rather than a continuous herbivory or grazing rate, the EHM model describes grazing as a series of episodes or periods of stay that each remove a fraction of vegetation biomass, followed by a time period of rest. I then use management inputs, specifically stocking density and period of stay, and the plant maximum relative growth rate [23], to define the influence of episodic grazing. Predicted changes in plant biomass drive aboveground and belowground production, which are then linked to a modified version of the SNAP soil carbon dynamic model [13]. The resulting model is expanded relative to [13] to include explicit broad climate conditions of mean annual temperature and precipitation and the influence of different dominant vegetation types, e.g., perennial grasses, annuals, and shrubs. The resulting model allows prediction of soil carbon dynamics as a function of grazing management decisions in many different environmental contexts. The model is then tested by comparing its predictions of soil organic carbon for pastures in central Montana, USA, subjected to different histories of grazing and land use management.

## 2. Methods

### 2.1. Episodic Herbivory Component of SNAPGRAZE Model

　　　I develop a model that links a previous model of soil carbon dynamics called SNAP [13] linked to the EHM [23] to generate a new model called SNAPGRAZE that accounts for management decisions on herd size and period of stay (days) in a pasture (defined grazing area), and the growing season length in days.

### 2.1.1. Grazing Model

　　　The influence of producer decisions on soil carbon is driven by their influence on aboveground plant production. The EHM calculates production by tracking biomass in a pasture due to plant growth in response to and following episodes of grazing. Producers decide the length of time prior to grazing, and the length of the period of stay, which includes one or more grazing events, and the length of the growing season constrain how much time forage plants can grow in the absence of grazing (Figure 1). Time decision variables are E the number of days prior to grazing, $D$ the period of stay or the number of days animals are present in a pasture, and $F$ the number of days in the growing season after grazing animals are removed—all of which add up to $G$, the number of days in the growing season. The $S$ terms refer to biomass at these different times in the grazed condition, starting with an early season initial biomass $S_0$ set by carbon stores in rhizomes, approximately $0.1\,S_K$ [24], $S_E$ biomass prior to grazing, $S_G$ biomass at the end of grazing, and $S_F$ biomass in grazed pasture at the end of the growing season. In the ungrazed case, biomass increases from $S_0$ to an ungrazed steady state $S_K$.

　　　Seasonal production in the absence of grazing is ultimately limited by resources such as water, soil nutrients, or light (self-shading), such that the relative growth rate, $RGR$ in g g$^{-1}$ day$^{-1}$, is density dependent, or declines with increasing biomass (g/m$^2$) in the absence of grazing [19,23]. Under these conditions, biomass accumulation over a time $t$ (days) is

$$S(t) = \frac{S_K S_0}{S_K e^{-rt} + S_0[1 - e^{-rt}]} \qquad (1)$$

where $r$ is the maximum relative growth rate (g.g$^{-1}$.day$^{-1}$) of the forage and $S_K$ is the steady state biomass in the absence of grazing (such as peak biomass in a grazing exclosure). Because plant growth

is assumed to be density (biomass) dependent [23], $r$ can be estimated as the intercept of a relationship between measured *RGR* and biomass [23,25] or can be estimated from growth experiments in pot studies with field soils [26–28]. Production in the absence of grazing, $P_U$, is then quantified as the increase in biomass from an initial amount $S_0$ to a biomass $S_K$ after $G$ days [23].

$$P_U = S_K - S_0 \tag{2}$$

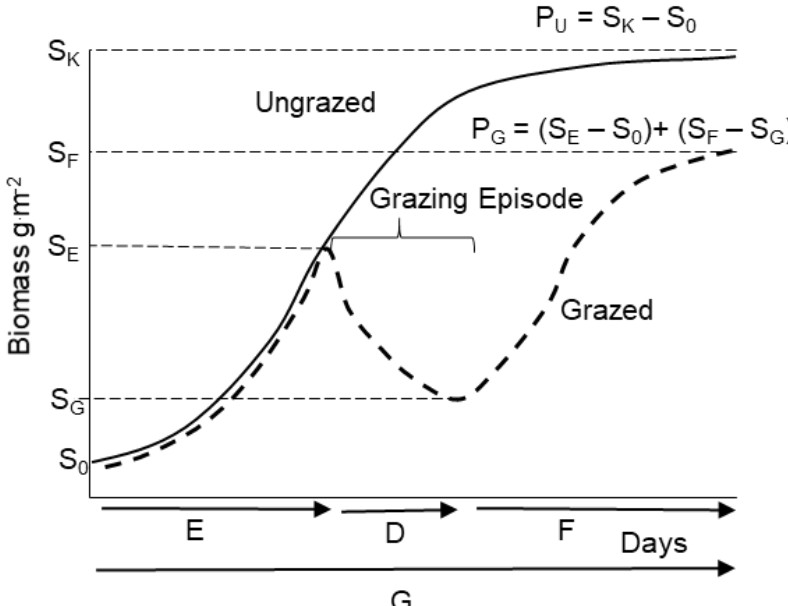

**Figure 1.** Theoretical average biomass accumulation over time under an ungrazed (solid curve) and episodically grazed (dashed curve) scenario. In ungrazed conditions, biomass accumulates from initial biomass $S_0$ to a maximum, $S_K$, leading to a production $P_U$.over the growing season $G$ days is The grazed period of stay begins $E$ days after the start of the growing season at initial biomass $S_E$ and lasts $D$ days, reducing biomass to $S_G$. Regrowth following grazing for $F$ days accumulates biomass to $S_F$ at the end of the growing season. Production under episodic grazing is $P_G$.

To determine grazed seasonal production, $P_G$, I assumed that individual pastures are grazed only $D$ days centered on the median day in the overall growing season of $G$ days, production is dictated by changes in forage biomass during three different time intervals (Figure 1). From the beginning of the growing season at initial biomass $S_0$, plant growth for $E$ days yields biomass $S_E$, a pre-grazing production

$$P_E = S_E - S_0 \tag{3}$$

where

$$S_E = \frac{S_K S_0}{S_K e^{-rE} + S_0 (1 - e^{-rE})} \tag{4}$$

Following this initial production period, $N$ animals that each consume $C_G$ plant biomass daily are introduced into some number $n$ of "pastures" of area $A$ (ha) with initial aboveground biomass $S_E$. Grazing during the period of stay $D$ days reduces the plant relative growth rate by $g$ (g g$^{-1}$·day$^{-1}$), the instantaneous per unit biomass loss rate due to grazing.Converting the rate of removal to a relative loss rate requires dividing consumption by biomass. Re-defining overall stocking density $d$ (animals/ha) as $N/A$ yields the rate of biomass loss during the period of stay, $L_G$

$$L_G = D * d * C_G * n * 10^{-4} \tag{5}$$

The factor $10^{-4}$ converts per hectare animal density into per m$^2$. While in reality biomass changes continuously over the period of stay, imposing plant biomass as a variable prevents an analytical solution to the calculation of biomass accumulation (or loss), so, as an approximation, I used $S_E$ the biomass at the beginning of the period of stay: $g = d * C_G * n * 10^{-4}/S_E$.

The amount of biomass at the end of the grazing episode is described by

$$S_G = \frac{S_K S_E}{S_K e^{-(r-g)D} + S_E\left[1 - e^{-(r-g)D}\right]} - L_G \tag{6}$$

where the left-hand term shows the biomass that would result from density dependent growth over $D$ days in the absence of grazers, starting at biomass $S_E$. The right-hand term is the total biomass removed by grazers over the $D$ days of the period of stay. At low stocking densities and less intense grazing, $S_G$ may be higher than $S_E$, indicating that biomass increases through the grazing episode. However, at typical production-level stocking densities, $S_G$ will be less than $S_E$, yielding the outcome depicted in Figure 1.

The term $C_G$ is daily consumption of biomass per animal, which is determined by animal body size $W$ (kg) under an assumption of digestive capacity limitation [29]. For ruminant livestock, I used Weckerly's [30] log-linear model $C_G = 5300 + 770 * \ln(W)$). However, active, lactating cows of a variety of breeds appear to have twice this daily intake [31], on average, and this multiplier likely applies to ungulates of all sizes [32–34]. Consequently, I used

$$C_G = 2 \times [5300 + 770 * \ln(W))] \tag{7}$$

Locally relevant per animal consumption rates can also be used.

Forage biomass at the end of the growing season, $S_F$, depends on biomass accumulation from the end of the period of stay at biomass $S_G$ until the end of the growing season over days $F = G - E - D$.

$$S_F = \frac{S_K S_G}{S_K e^{-rF} + S_G[1 - e^{-rF}]} \tag{8}$$

Under the assumption that $S_G < S_E$

$$P_F = S_F - S_G \tag{9}$$

total seasonal production under the grazing management system, $P_G$, is the sum of $P_E$ plus production after the grazing episode $P_F$ [23] (Figure 1).

$$P_G = (S_E - S_0) + (S_F - S_G) \tag{10}$$

When grazing is light enough that plant growth exceeds forage consumed, $S_G > S_E$, and

$$P_G = (S_E - S_0) + (S_F - S_E) \tag{11}$$

Plant biomass also undergoes dynamics during the dormant season in ways that may affect SOC. Biomass declines over the dormant period 365-$G$ days from $S_F$ by a total dormant season biomass loss $L_O$ (g/m$^2$) at stocking density $d \times 10^{-4}$ and per animal consumption $C_G/2$. Per capita consumption is divided by 2 to account for lower per animal consumption to just maintain weight rather than gain fat or lactate calves during the dormant season [29–31]. Period of stay and pasture number are unimportant because there is no re-growth; loss is the same whether consumption occurs by many livestock over a few days or the system stocking density over the entire dormant season.

$$L_O = (C_G/2)(365-G)d \times 10^{-4} \tag{12}$$

Finally, livestock grazing must be sustainable—that is, a given stocking density and grazing scheme combination must be feasible. Loss of biomass during the offseason from the final biomass at the end of the growing season, $S_F$, was judged to be unsustainable if it reduced biomass below $S_0$, or if $S_F - L_O < 0.1 S_K$. Substituting Equation (12) for $L_O$ and solving for $d$ yields a maximum stocking density

$$d_{max} = [(S_F - 0.1 S_K) \times 10^4]/[(C_G/2)(365 - G)] \tag{13}$$

Variables in the SNAPGRAZE model are presented in Table 1.

The EHM component of SNAPGRAZE was analyzed to determine how decreasing periods of stay and increasing overall system and local pasture stocking rates would affect forage production. To assess the effect of climate on these outcomes, I systematically varied mean annual temperature (*MAT*) and mean annual precipitation (*MAP*) and explored two values for *r* based on reported average values of 0.035 for a temperate grassland system [25] and 0.05 for a tropical system [23].

### 2.1.2. Soil Carbon Dynamics

Changes in SOC produced by different grazing management schemes were determined with a modification of the SNAP model otherwise developed for migratory grazing systems in the Serengeti ecosystem [13]. The original SNAP model tracks the fate of recalcitrant carbon (lignin and cellulose) fixed during production (both above and belowground) and its potential conversion to either soil organic matter or $CO_2$. The original SNAP model derived from data from the Serengeti ecosystem that was applicable to tropical grasslands and savannas. The original SNAP model determined production from just rainfall and water holding capacity, as dictated by soil sand content. Serengeti is heavily dominated by perennial grasses, which have relatively high lignin and cellulose content. Grazing removed biomass and had non-linear effects on production—of which, the lignin and cellulose proportion from both shoots and roots entered the soil carbon pool as plant-derived soil organic carbon in year $y$ ($PDSOC_y$). A proportion of lignin and cellulose in consumed biomass is returned to the soil as dung-derived SOC in year $y$ ($DDSOC_y$). These inputs are then converted back into $CO_2$ through annual microbial respiration in year $y$ ($MRESP_y$) over some number of days with sufficient soil moisture to support microbial activity (*WETDAYS*). Thus, the change in SOC in a given year is

$$\Delta SOC_y = PDSOC_y + DDSOC_y - MRESP_y \tag{14}$$

Functions for these sources and losses of *SOC* were originally determined from Serengeti datasets [35,36] using regression equations and logical process models [13]. Here, I modify these functions to produce a new model called SNAPGRAZE. This new model accounts for a more general set of conditions that includes temperate grasslands and shrublands, the effect of temperature on production, environments dominated by shrubs and annual plants instead of just perennials, and different growing season lengths. In addition, productivity estimates are driven by the explicit set of grazing management decision variables and are linked to the biomass and production estimates of the EHM component of SNAPGRAZE in Section 2.1.1. These new functions are summarized below, with most coefficients derived from the original SNAP model [13].

**Table 1.** Variables, definitions and units in alphabetical order for the SNAPGRAZE model of forage production and soil carbon dynamics.

| Parameters | Definition | Units | Determination |
|---|---|---|---|
| $A$ | Total area of grazing system | $m^2$ | Input |
| $ANPP_{max}$ | Annual aboveground production in absence of grazing | $g\ m^{-2}$ | $12.04 - 25.18/(0.0083 * (MAT + 273.15)) + 0.72 * \ln(MAP)$ |
| $ANPP_{est}$ | Annual aboveground production under grazing | $g\ m^{-2}$ | $(S_E - S_0) + (S_F - S_E)$; $P_G = ANPP_{est}$ |
| $BNPP$ | Annual belowground productivity under grazing | $g\ m^{-2}$ | $[0.602 * MAP - 0.00038 * MAP^2 + 5.88 * MAT] * (P_G/P_U) * APC * (DEPTH/40)$ |
| $C_G$ | Per animal daily consumption | $g\ animal^{-1}\ day^{-1}$ | $2 * [5300 + 770 * \ln(W))]$ or Input |
| $D$ | Period of stay | days | Input |
| $d$ | Stocking density | Animals $ha^{-1}$ | $(N/A) * 10^{-4}$ |
| $DDSOC_y$ | Dung-derived soil carbon input in year $y$ | $gC\ m^{-2}\ year^{-1}$ | $LIGCELL * 0.45 * 0.3 * (DC_G nd + L_O)$ |
| $DEPTH$ | Soil sampling depth | cm | Input (typically 30 cm) |
| $DMRESP$ | Daily microbial respiration | $gC\ m^{-2}\ day^{-1}$ | $-0.579 + 0.00044 * SOC$ for SOC > 4600 g $m^{-2}$; OR $\exp(10.18) * SOC_t^{1.298}$ for SOC < 4600 g $m^{-2}$ |
| $E$ | Time prior to grazing episode | days | Input |
| $F$ | Days from end of grazing period of stay to end of the growing season, $G - (E+D)$ | days | $G - (E+D)$ |
| $g$ | Relative loss rate of biomass to grazing | $g\ g^{-1}\ day^{-1}$ | $d * C_G * n * 10^{-4}/S_E$ |
| $G$ | Length of plant growing season | days | $22.99 * MAT - 0.94 * MAT^2 + 0.073 * MAP$ |
| $L_G$ | Biomass removed during grazing period of stay | $g\ m^{-2}$ | $D * d * C_G * n * 10^{-4}$ |
| $L_O$ | Biomass removed during the dormant season | $g\ m^{-2}$ | $(C_G/2)(365 - G)d * 10^{-4}$ |
| $LIGCELL$ | Mean proportion of plant as lignin and cellulose | % | Input |
| $MAP$ | Mean annual precipitation | mm | Input |
| $MAT$ | Mean annual temperature | $^oC$ | Input |
| $MRESP_y$ | Microbial respiration rate in year $y$ | $gC\ m^{-2}\ year^{-1}$ | $WETDAYS * (0.7+0.3 * SAND\%/100) * DMRESP$ |
| $N$ | Number of animals in grazing system | | Input |
| $n$ | Number of pastures in grazing system | | Input |
| $PDSOC_y$ | Plant-derived soil carbon input in year $y$ | $gC\ m^{-2}\ year^{-1}$ | $0.45 * [\ LIGCELL * (S_F - L_O/2) - (1 - FIRE) + (LIGCELL + 0.05) * BNPP]$ |
| $r$ | Maximum relative growth rate | $g\ g^{-1}\ day^{-1}$ | Input |
| $S_0$ | Biomass at the onset of the growing season, produced from resource reserves | $g\ m^{-2}$ | $0.1 * S_K$ |
| $S_E$ | Biomass at start of grazing episode | $g\ m^{-2}$ | $\frac{S_K S_0}{S_K e^{-rE} + S_0[1 - e^{-rE}]}$ |
| $S_F$ | Biomass at the end of the growing season | $g\ m^{-2}$ | $\frac{S_K S_G}{S_K e^{-rF} + S_G[1 - e^{-rF}]}$ |
| $S_G$ | Biomass at the end of the grazing episode | $g\ m^{-2}$ | $\frac{S_K S_E}{S_K e^{-(r-g)D} + S_E[1 - e^{-(r-g)D}]} - L_G$ |
| $S_K$ | Biomass in absence of grazing | $g\ m^{-2}$ | $ANPP_{max}/0.9$ |
| $SAND\%$ | Percent soil as sand | % | Input |
| $\Delta SOC_y$ | Change in soil carbon density in year $y$ | $g\ m^{-2}\ year^{-1}$ | $PDSOC_t + DDSOC_t - MRESP_t$ |
| $W$ | Mean mass of grazing animals | kg | Input |
| $WETDAYS$ | Days soil moisture > 10% for microbial activity | days | $(0.00044 * RAIN - 0.025) * G$ |

### 2.2. Literature Review and Analysis of Climate Association with Production

Values of aboveground net primary production (*ANPP*) for grasslands were obtained from the ORNL DAAC database), featuring 37 sites with reported ANPP over multiple years along with *MAP* (mm) and *MAT* (°C) [37]. To this database, I added reported *ANPP* from an additional seven tropical grassland sites [36]. For *BNPP*, the Web of Science® keyword phrase (grassland* and temperature and (precipitation or rainfall) and ("root production" or "root productivity" or "belowground production" or "belowground productivity") yielded 20 results. Eight of these provided data from one or more years and/or sites to yield 36 estimates of *BNPP* associated with *MAT* and *MAP* [38–46]. A similar literature search was conducted on Web of Science® using keyword phrase (grassland* and temperature and (precipitation or rainfall) and ("length of growing season" or "growing season length"), which yielded 61 results—nine of which yielded a total of 39 estimates of growing season length (G, days) along with MAP and *MAT*.

### 2.3. Soil and Vegetation Sampling

A soil and vegetation survey was conducted in the period June 25–30, 2017 at 31 sampling stations in irrigated and non-irrigated pastures across nine private ranches in south-central Montana (Figure 2). Soils were predominantly clay loams with varying degrees of siltstone cobble. There were two sampling stations in each of (a) native pasture previously continuously grazed, (b) native pasture grazed as part of an *n* pasture system for a known number of years, and (c) irrigated pasture (mostly formerly hayed but now grazed as part of the rotational system). Some ranches no longer had continuously grazed pastures, and some were not grazing irrigated land. Ranch owners or managers were interviewed to determine pasture grazing history prior to sampling.

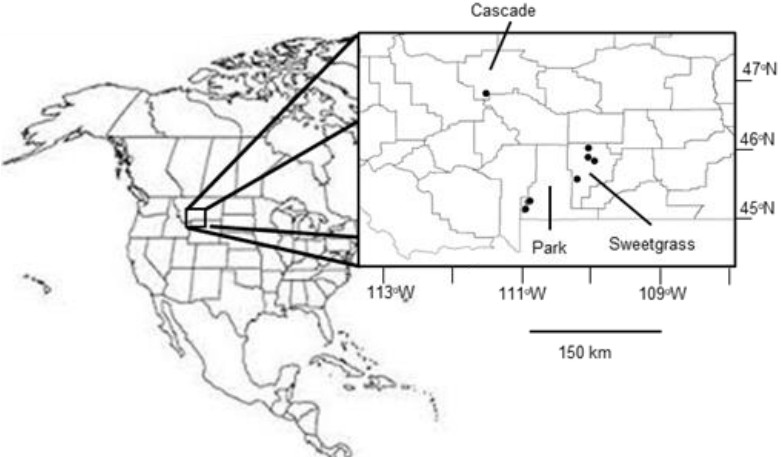

**Figure 2.** Map of approximate location and counties of the nine Montana, USA, ranches sampled to test the SNAPGRAZE model of grazing and soil carbon dynamics. Note four ranches are clustered in the apparent two southernmost points.

Soil for carbon analysis was sampled from two cores with an auger of a diameter of 7.6 cm to a depth of 20 cm. The two auger samples were then mixed in a tub; a 100 g sample was then extracted and placed in a sealed plastic bag. Clay content (*CLAY*%) was estimated byusing the standard field ball and soil ribbon method to estimate the soil classification (e.g., clay loam, sandy loam), following a published decision tree [47]. Clay content was read as the midpoint clay content associated with the soil class from a standard soil classification triangle [48]. This sample was later dried at 105 °C before analysis of SOC% by loss on ignition (*LOI*) at 550 °C for 24 h. Carbon content was estimated as *SOC*% = 0.56 (*LOI* − 0.075 * *CLAY*%) [49]. A depth of only 20 cm was used because of a persistent skeletal subsoil below 15 cm at many sampling stations.

Bulk density (mass of soil per unit volume) was measured by pounding in a steel pipe with a diameter of 5 cm to a depth of 20 cm, removing soil around one side of the pipe, sliding a metal plate underneath the pipe and then lifting out the sample in the pipe. Soil was sieved through a mesh screen with a pore size of a 2 mm to remove rocks whose volume was later measured by water displacement. Sieved soil from the known core volume was dried at 105 °C and weighed. Bulk density of fine soil ($BD$, g soil/cm³ soil) was determined by dividing soil dry mass by fine soil volume (core volume—rock volume) [50]. Soil organic carbon ($SOC$) density (Mg ha$^{-1}$) was determined by $SOC = SOC\% * BD * DEPTH * (1 - V_R/V_T)$, where $V_R$ = rock volume (cm³) and $V_T$ = total core volume (cm³) and DEPTH is in cm, following Equation (5) in [50].

Dominant vegetation was also surveyed as the visual percent cover of four functional groups, shrubs, perennial grasses, annual grasses, and forbs, in a circle with a radius of 8 m surrounding the location of soil cores. A plot center was determined as the GPS point, and a rope 8 m in length was extended from a stake at the center. Small plastic cones were placed along the perimeter of the circle at five pace intervals. A visual assessment of the percent cover of each plant functional group was made within each of the four quarters of the circle. The assigned dominant plant group for the plot was that with the highest mean cover among the four quarters.

*2.4. Statistical Analysis*

The model inputs for *ANPPmax*, *BNPP* and *G* were determined from multiple regressions of these variables (as dependent variables) with *MAP* and *MAT* (independent variables). Univariate relationships were inspected to determine whether they were linear or non-linear. All three dependent variables were initially included in a linear regression model without transformed independent variables (linear model). The most appropriate non-linear transformations were then applied in a second regression (non-linear model). The two models were compared on the basis of adjusted $R^2$ and on whether the coefficients of either model were significantly different from zero ($P < 0.05$). For *ANPP*, *ANPP* and *MAP* were ln-transformed and *MAT* was converted to °K and inverted as $1/(R * (MAT + 273.15))$ to generate an Arrhenius relationship (where $R$ is the gas constant 0.00834 kJ mol$^{-1}$ K$^{-1}$). This i a standard approach for assessing biological responses to temperature [51]. For *BNPP*, univariate relationships demonstrated a unimodal pattern versus precipitation, likely related to the allocation by plants of fixed carbon to stems at higher rainfall, presumably to avoid light competition. Consequently, the non-linear model for *BNPP* included a quadratic term for precipitation. Finally, for growing season length, *G*, univariate relationships revealed a unimodal pattern versus *MAT*, presumably reflecting the shortening of *G*, for a given precipitation, by dry conditions at higher temperatures. Consequently, the non-linear model for *G* included a quadratic term for *MAT*. In addition, in the case of BNPP and *G*, the intercept was set at zero so that any regression model could not yield negative values for these two inputs, which could produce unrealistic and invalid predictions in the SNAP soil carbon model.

Soil organic carbon (SOC) densities were predicted for different pastures with different land use and grazing histories. Predictions were based on local data for *r*, mean percent of plant as lignin and cellulose (*LIGCELL*), *MAP*, *MAT*, percent soil as sand (*SAND%*), *N*, *n*, and *D* for different numbers of years in the past. For example, a pasture might have been seasonally grazed (*D* > 90 days) from 1900 to 1976, as part of a 4-pasture rotation (*D* = 40 days) from 1976 to 2008, and as part of an 8-pasture rotation (*D* = 20 days) from 2008 to 2017. Annual change in SOC density ($\Delta SOC$) for this example was then calculated on the basis of change from an original baseline equilibrium *SOCeq* for continuous grazing followed by $\Delta SOC$ for 4-pasture conditions for 32 years and for 8-pasture rotation for 9 years, yielding an expected SOC as of 2017. To evaluate correspondence of predicted values to measured SOC values, I used linear regression and determined an overall slope, intercept, and $R^2$ for all 24 sites.

## 3. Results

### 3.1. Associations of Climate and Seasonal Production

Substantial variation in *ANPP*, *BNPP* and *G* was explained by *MAT* and *MAP*. Non-linear models fit the data better than linear models for all three dependent variables, as $R^2_{adj}$ was > 0.2 higher than for linear models, but the form of the multi-variable model was different in each case. The non-linear regression models, as justified by the univariate relationships in Figure 3, are presented in Table 2.

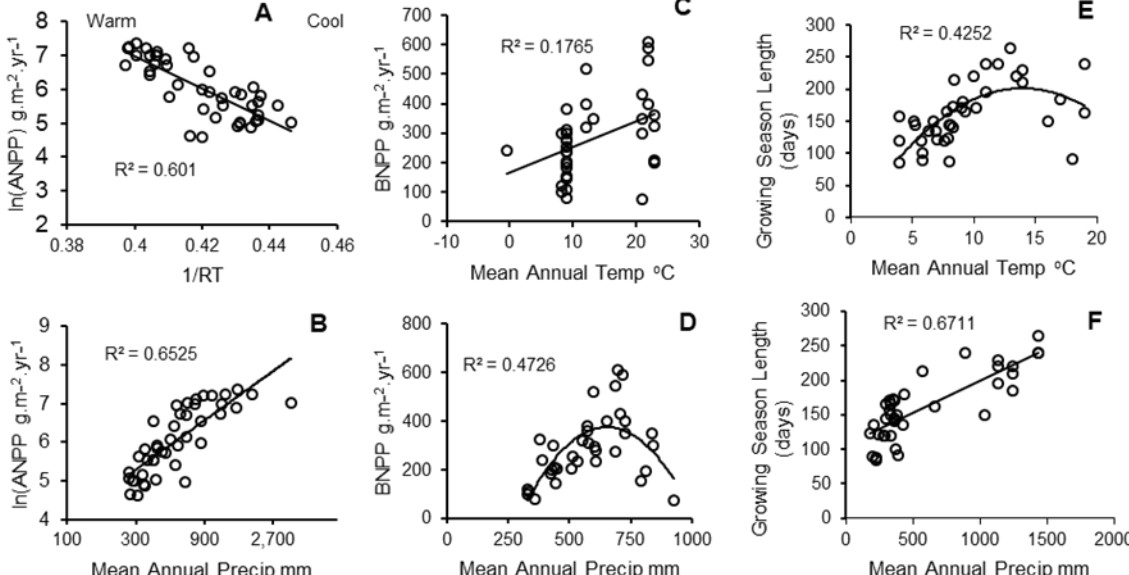

**Figure 3.** Univariate relationships between three driving variables for soil organic carbon versus each of two climate variables, mean annual temperature (*MAT*, upper row) and mean annual precipitation (*MAP*, lower row): (**A**,**B**), aboveground net primary production (*ANPP*); (**C**,**D**), belowground net primary production (*BNPP*); (**E**,**F**), annual length of the growing season (*G*). Note the logarithmic scale for mean annual precipitation in (**B**). Non-linear relationships in (**D**,**E**) are quadratic relationships that fit significantly better than linear relationships (improvement in $R^2$ > 0.2, reduction in Akaike Information Criterion (AIC) > 10).

**Table 2.** Coefficients of mean annual temperature (MAT) and mean annual precipitation (MAP) generated from multiple regressions of three independent variables—logarithm of aboveground production ln(ANPP), belowground production BNPP, and growing season length, G.

| | Dependent Variable | | | | | |
|---|---|---|---|---|---|---|
| | **ln(*ANPP*)** | | ***BNPP*** | | ***G*** | |
| **Parameter** | **Coefficient** | **SE** | **Coefficient** [1] | **SE** | **Coefficient** | **SE** |
| Intercept | 12.04 | 3.53 | 0 | N/A | 0 | N/A |
| *MAT* | | | 5.89 | 3.12 | 22.99 | 1.73 |
| *MAT*$^2$ | | | | | −0.94 | −0.1 |
| 1/R(MAT+273.15) | −25.18 | 6.76 | | | | |
| *MAP* | | | 0.602 | 0.183 | 0.073 | 0.014 |
| *MAP*$^2$ | | | −0.00038 | 0.00022 | | |
| ln(*MAP*) | 0.718 | 0.145 | | | | |
| *N* | 44 | | 36 | | 38 | |
| $R^2_{adj}$ | 0.733 | | 0.835 | | 0.944 | |

[1] Regressions for *BNPP* and *G* were performed with the intercept set to 0 to avoid the possibility that the regression equation would yield negative values that cannot occur in nature.

### 3.1.1. Grazing Management and Forage Production

EHM Component of SNAPGRAZE

The statistical models for *ANPPmax*, *BNPP* and *G* plus the determination of *ANPPest* (which = $P_G$) (Equations (10) and (11)) were inserted into the original model to produce the SNAPGRAZE model. Maximum aboveground (*ANPPmax*) and seasonal belowground production (*BNPP*) in the absence of grazing were estimated from two climate variables: *MAT* and *MAP*, based on literature reviews.

Estimated seasonal production under grazing (*ANPP_{est}*) was determined from the EHM component: Equations (4), (8), and (9) or (10), using the parameter *r*, the steady state biomass in the absence of herbivory, $S_K$, and the initial biomass at the beginning of the season $S_0$ (established from carbohydrate and nutrient reserves in rhizomes or seeds, estimated as 10% of $S_K$). $S_K$ was estimated as a function of climate and soil texture: $S_K = ANPP_{max}/0.9$ and

$$ANPP_{max} = e^{[12.039 + 0.718 * \ln(PRECIP) - (25.18/(R * (273.15+MAT)))]} * (1.33 - 0.0075 * SAND\%) \qquad (15)$$

where *R* is the gas constant 0.00834 kJ mol/K. The exponential form of the function accounts for a non-linear relationship between production and temperature (converted to linear by an Arrhenius function [51]) and a double logarithmic relationship with *MAT*. The expression (1.33 − 0.0075 * *SAND%*) from [13] weights the production estimate by soil water holding capacity, which decreases with increasing soil sand content.

### 3.1.2. Model Analysis

The effects of increasing pasture number and de facto decreasing periods of stay at a given system-wide stocking density were similar across cool to warm and dry to wet climates (Figure 4). For a given system-wide stocking density, production generally increased by over 200% in shifting from a continuous or 2-pasture rotation to an 8- to 16-pasture rotation. However, climate mattered considerably for the magnitude of these effects and the likelihood that grazing would stimulate production.

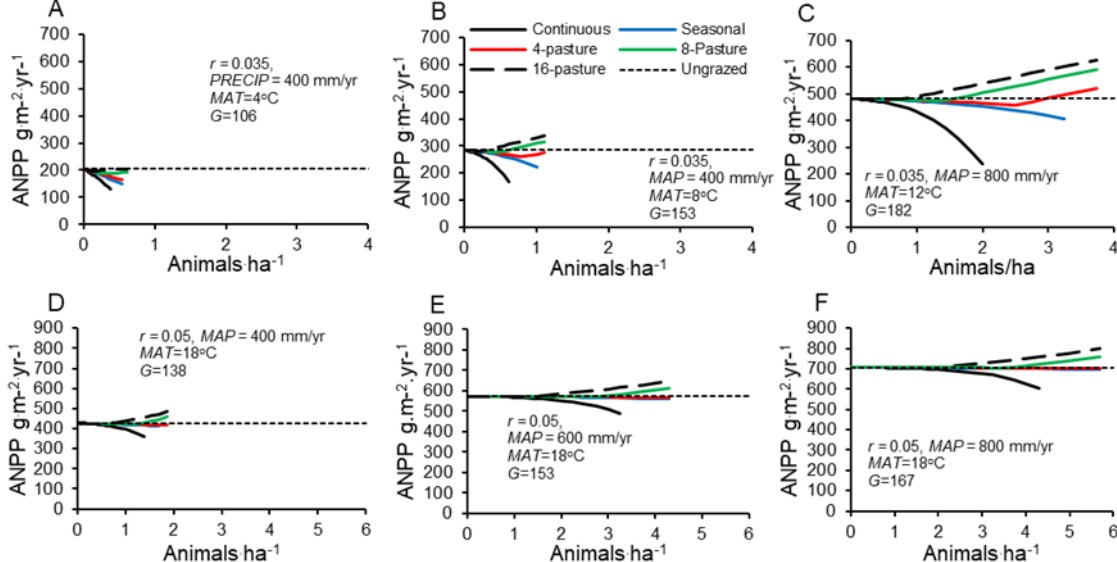

**Figure 4.** Grassland seasonal production under different grazing schemes, represented by the number of pastures animals are moved through (Continuous = 1, Seasonal = 2) during a plant growing season and different stocking densities for the entire grazing system. (**A–C**): Temperate climates; (**D–F**): Tropical climates. Production is calculated for different climates, including mean annual temperature (*MAT*) and precipitation (*MAP*) and growing season lengths, *G* and their associated maximum relative growth rates, *r*.

### 3.2. Soil Carbon Dynamics Model

Belowground production, $BNPP_{est}$, shows a unimodal response to *MAP* and linear increase with MAT (Figure 3C,D, Table 2) and I modify the function further to scale *BNPP* to *ANPP* and apply a correction for the dominance of annual versus perennial grasses.

$$BNPP = [0.602 * MAP - 0.00038 * MAP^2 + 5.88 * MAT] * (P_G/P_U) * APC * (DEPTH/40) \quad (16)$$

The expression for *BNPP* as a function of rainfall results from regression analysis of annual root production in grasslands that varied in *MAT* (see Section 2.2). The factor $P_G/P_U$ adjusts the estimate for the effects of grazing on production and thus carbon assimilation allocated to roots. The factor *APC* is a factor that that corrects for the influence of annual versus perennial plant growth strategies on belowground production. Perennials dominate the Serengeti system used to derive the original SNAP model, but annual grasses allocate much less biomass belowground and may be locally dominant, particularly in degraded grasslands. Based on the average ratio of *BNPP* to *ANPP* for annual plants compared to perennials in a limited literature review [52–54], *APC* = 0.291 if forage is dominated by annuals or shrubs that are often associated with annuals in drier grasslands. Otherwise *APC* = 1. Finally, the original belowground production values were measured to 40 cm depth in the Serengeti, and the factor *DEPTH*/40 corrects for different depths at which *BNPP* might be estimated.

Plant-derived *SOC* in year *y*, $PDSOC_y$, is determined by the proportion of plant biomass as lignin and cellulose (*LIGCELL*) of both aboveground and belowground production. Aboveground production remaining at the end of the growing season, $S_F$, may be lost to fire (*FIRE* = proportion of previous 10 years in which fires occurred) and lignin and cellulose content of roots was assumed to be 5% higher than for aboveground plant tissue [13]. Proportion of lignin and cellulose as carbon [13] was assumed to be 0.45.

$$PDSOC_y = 0.45 * [ LIGCELL * (S_F - L_O/2) * (1 - FIRE) + (LIGCELL + 0.05) * BNPP] \quad (17)$$

The difference, $S_F - L_O/2$, is the mean standing biomass available for decomposition over the dormant season, as the date of consumption could occur early or late and on average occurs halfway through the dormant season.

Soil organic carbon is also derived from dung deposited in year *y* ($DDSOC_y$), as described in detail in [13]. This calculation of DDSOC is modified here to use the parameters driving biomass removed by grazers, $L_G$ from Equation (5), and 0.3 is a factor accounting for the fraction of consumed mass that remains as excreted dung.

$$DDSOC_y = LIGCELL * 0.45 * 0.3 * (DC_g nd + L_O) \quad (18)$$

Finally, carbon losses associated with microbial respiration in year *y* ($MRESP_y$) were accounted for as a function of *WETDAYS*, the number of days in which soil moisture exceeds a threshold of 10% for microbial activity [13]. The days soil moisture exceeded a threshold was scaled to local climate by multiplying by growing season length, *G*.

$$WETDAYS = (0.00044 * MAP\text{-}0.025) * G \quad (19)$$

The maximum daily rate of microbial respiration (*DMRESP*) is determined by a stepwise function, as determined by lab incubations of Serengeti soils [13,35]. This function was based on a linear regression of respiration versus soil organic carbon data derived from [35], evaluated over SOC densities from 4000 to 10,000 g/m$^2$ to a depth of 40 cm.

$$DMRESP = -0.579 + 0.00044 * SOC_y \quad (20)$$

The negative intercept of this function implies that carbon respiration by microbes is negative at low soil carbon values, which is logically impossible. To better account for microbial respiration at low soil carbon densities, we fit a second, non-linear function to the same data (Figure 5)

$$\ln(DMRESP) = -10.872 + 1.296 \ln(SOC_y) \tag{21}$$

which can be re-written as the power function.

$$DMRESP = e^{(-10.18)} * SOC_t{}^{1.298} \tag{22}$$

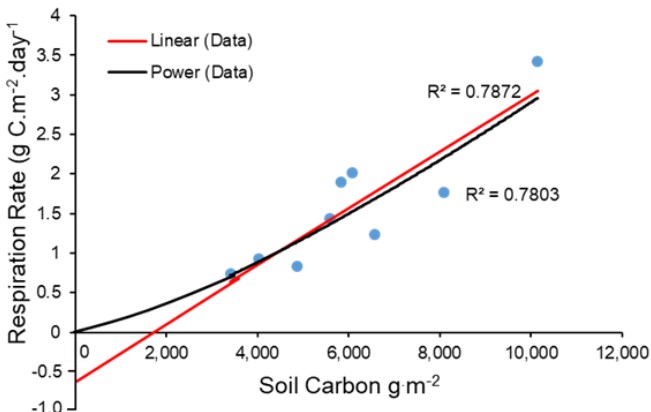

**Figure 5.** Two alternative regression lines fit to measures of microbial respiration plotted versus measured soil carbon density (data from [35]). The red line is a linear fit, while the black curve is a power function fit from a ln-ln regression. Note that extrapolation of the linear fit to low soil carbon densities results in untenable negative estimates of microbial respiration.

Overall average total annual microbial respiration in year $y$ ($MRESP_y$) for project and baseline calculations of SOC were then based on a piecewise function for daily respiration in which the exponential function is applied at low SOC (approximately $< 4600$ g/m$^2$) and the linear function applied at higher SOC. Annual microbial respiration is then calculated

$$MRESP_y = WETDAYS * (0.7 + 0.3 * SAND\%/100) * DMRESP \tag{23}$$

where the expression (0.7 + 0.3 * $SAND\%$/100) corrects the respiration rate for greater microbe accessibility to SOC in sandy soils [35].

These equations account for all the major inputs and losses of stable C into soil. Using Equation (14) and substituting Equations (17), (18), and (23) yields the change in SOC in year $y$

$$\Delta SOC_y = PDSOC_y + DDSOC_y - MRESP_y \tag{24}$$

By setting $\Delta SOCt = 0$, the above equation can be solved for the $SOC_y$ term in $MRESP_y$ to yield an equilibrium $SOC_{eq}$, depending on the appropriate choice of function for $DMRESP$ (Equations (20) or (22)).

$$SOC_{eq} = \frac{PDSOC_y + DDSOC_y}{0.0044 * WETDAYS * \left(0.7 + 0.3 * \frac{SAND\%}{100}\right)} + 0.579 \text{ for SOC } > 4600 \text{ g/m}^2 \tag{25}$$

Or

$$SOC_{eq} = \left\{ \frac{PDSOC_y + DDSOC_y}{WETDAYS * \left(0.7 + 0.3 * \frac{SAND\%}{100}\right) * e^{-10.18}} \right\}^{\frac{1}{1.296}} \text{ for SOC } < 4600 \text{ g/m}^2 \tag{26}$$

### 3.3. Analysis of the Grazing Management Model

The regressions from Table 2 to predict *ANPPmax* and *G* were used in the EHM component of SNAPGRAZE, and several interesting outcomes emerged. In all simulations, grazing systems were viable if they met requirements of no loss of animal weight or reduction in forage biomass below $S_0$, the presumed plant production from belowground carbon reserves. Systems were viable across a range of stocking densities. The highest viable stocking densities were set by biomass remaining at the end of the growing season. The importance of increasing the number of pastures and decreasing period of stay was amplified at shorter growing seasons.

The range of feasible stocking densities generally increased with *MAT* and *MAP* and to a lesser extent with the number of pastures (shortness of periods of stay), such that doubling precipitation and temperature increased the range of stocking densities by approximately a factor of 10 (Figure 4). In temperate climates with continuous, seasonal and 4-pasture rotational grazing, all stocking densities decreased production relative to ungrazed conditions. Grazing enhanced production only at high precipitation with short-duration, high stocking density schemes.

In tropical climates (*MAT* = 18 °C) and accompanying higher *r* (=0.05) [23], grassland supported much higher stocking rates and grazing enhanced production over a much wider range of precipitation and pastures per system (Figure 4). In all simulated tropical climates, systems with four or more pastures showed no decline in production relative to ungrazed conditions across the entire range of stocking densities. In tighter rotation systems (*n* ≥ 8 pastures) with high stocking density, forage production increased to 15% above that in ungrazed systems (Figure 4D–F).

The effects of *r* and *MAT*, independently of grazing scheme (8-pasture rotation) and MAP (400 mm/year), were substantial. Increasing *r* by the difference between reported temperate climate values [22] and tropical climate values [23] led to a very large increase in production and stocking density (Figure 6A). Likewise, holding *r* constant (*r* = 0.035) and varying *MAT* yielded a dramatic increase in production and stocking density when *MAT* increased from 4 to 12 °C. However, a further increase in *MAT* had relatively little effect on sustainable stocking density and forage production (Figure 6B).

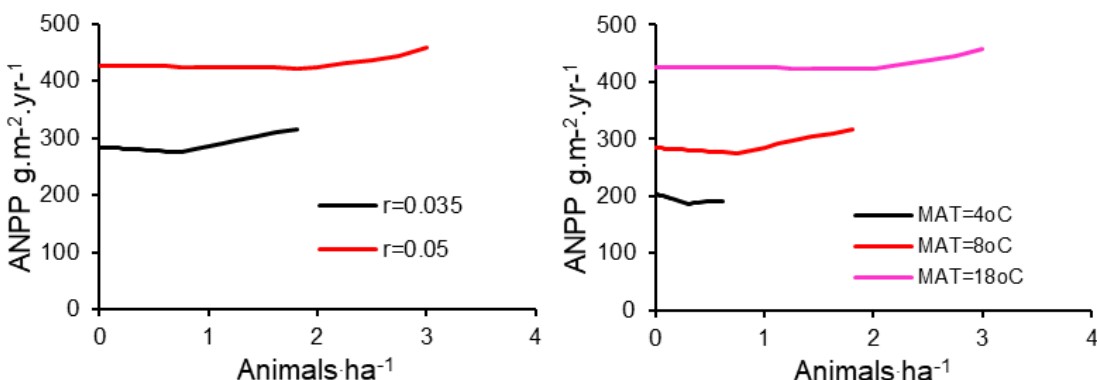

**Figure 6.** Grassland aboveground net primary production (*ANPP*) under 8-pasture grazing management, with mean annual precipitation = 400 mm, for different stocking densities for the entire grazing system. Production is calculated for (**A**) two different plant community maximum relative growth rates (*r*) and (**B**) three different mean annual temperatures (*MAT*).

### 3.4. Soil Carbon and Different Grazing Schemes

Overall, the model predicts improvement of soil carbon across a wide range of stocking densities and climates. Episodic grazing with sufficiently short periods of stay in sufficiently long growing season lengths can increase SOC above that of ungrazed conditions. The effects of grazing management on equilibrium soil carbon (Figure 7) had effects that largely mirrored those on forage production (Figure 4), except that grazers were less likely to have negative effects on carbon. Increases in temperature and

precipitation yield increases in growing season length, *G*, which allow shorter periods of stay on more pastures with higher but still feasible livestock stocking densities to increase soil organic carbon relative to ungrazed conditions by as much as 50–80% (Figure 7B–F). Increasing the number of pastures provides a diminishing return in additional soil carbon: the largest increase in soil carbon, particularly at sites with longer growing seasons, occurred at shifting from continuous to 2 and to 4 pastures, with much smaller increases occurring at shifts to 8 and 16 pastures.

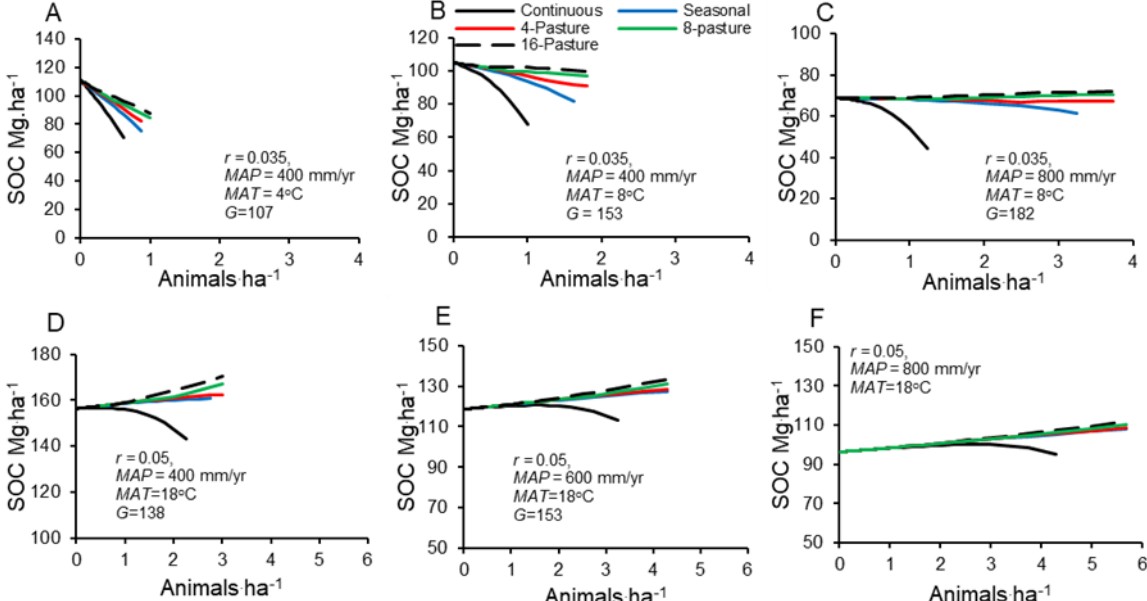

**Figure 7.** Equilibrium soil organic carbon densities (Mg ha$^{-1}$) to a depth of 30 cm for different numbers of pastures and stocking densities for the entire grazing system. (**A**–**C**), temperate conditions. (**D**–**F**), tropical conditions. Note the difference in scales for (**A**–**C**) versus (**D**–**F**).

Soil carbon densities are most sensitive to different grazing management schemes, in terms of percent improvement in soil carbon, in less productive systems with shorter growing seasons (dry or cold) (Figure 7A,D, Figure 8). At a given stocking density, increasing the number of pastures improves soil carbon proportionately more under cooler and/or drier conditions in both temperate and tropical temperature ranges (Figure 8).

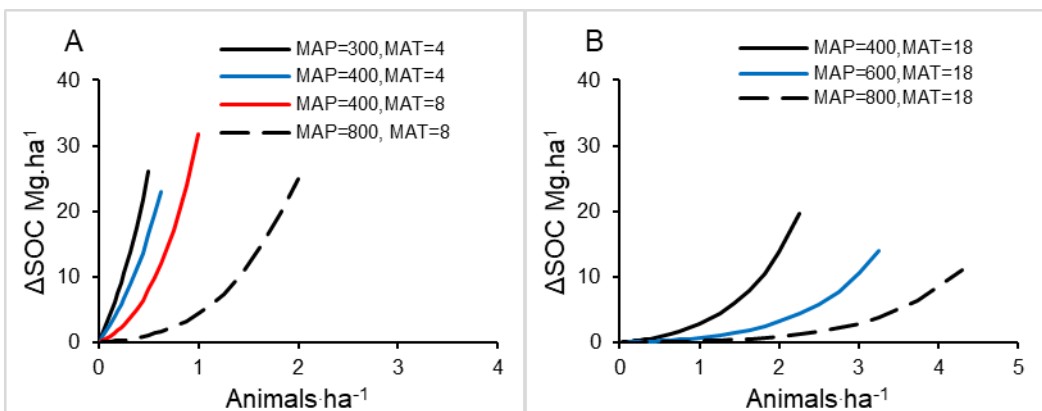

**Figure 8.** Potential change in equilibrium SOC (Δ*SOC*, Mg ha$^{-1}$) by switching from continuous grazing management to an 8-pasture rotation at a given stocking density under varying mean annual precipitation (*MAP*) and temperature (*MAT*) in two contexts: (**A**), temperate conditions and (**B**), tropical conditions.

### 3.5. Model Validation

To validate the model in real grazing systems, we gathered information about history of grazing systems and then measured soil carbon and vegetation characteristics and grazing management at 24 sampling stations in south central Montana (Figure 9, Table 3). Observed SOC density ranged from near 20 g/m$^2$ (20 cm depth) to over 120 g m$^{-2}$ as influenced by whether dominant plants were perennial grasses or sagebrush (*Artemisia tridentata*), length of time the pasture had been at a short-duration, high stocking density grazing scheme, and type of livestock (horses, bulls, cow-calf pairs). Observed SOC density to 20 cm corresponded well to predicted SOC density given the different grazing histories and vegetation covers encountered ($R^2$ = 0.816). The regression line between the two had a slope not significantly different from 1 ($p$ = 0.18) and an intercept not significantly different from zero ($p$ = 0.25).

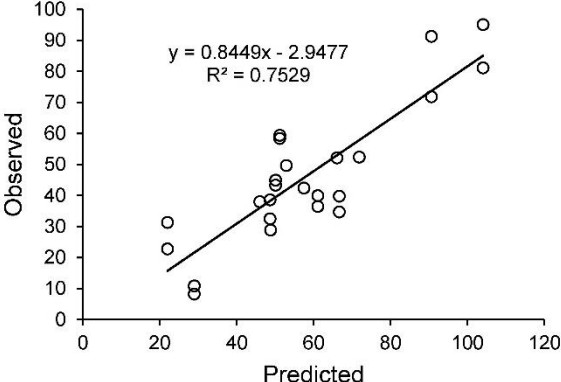

**Figure 9.** Observed soil organic carbon (SOC) density (Mg ha$^{-1}$ to a depth of 20 cm) versus predicted SOC from SNAPGRAZE for 24 sampling stations at the nine ranches in Montana, USA. Predicted values based on local inputs for *MAT*, *MAP*, and *LIGCELL* parameters and ranch grazing histories. Regression line is observed SOC = −2.25 (+ 6.39 s.e.m) + 0.85 (+ 0.10 s.e.m.) × predicted SOC, $R^2$ = 0.752, $p$ < 0.001.

**Table 3.** Grazing histories and resulting model parameters * for sampling stations at nine Montana, USA, ranches (two stations at each site).

| Ranch | Note | Vegetation | MAT °C | MAP mm/year | SAND% |
|---|---|---|---|---|---|
| A | Calving, horses at 0.37 animals ha$^{-1}$. | sage *** | 6.8 | 374 | 40 |
| B | Horses at 0.24 ha$^{-1}$, continuous since 1960. | perennials | 8.25 | 418 | 45 |
| B | Seasonal (2-pasture rotation) until 2002 at regional stocking densities **, 8-pasture rotation for 15 years, 0.26 cattle ha$^{-1}$. | perennials | 8.25 | 418 | 40 |
| C | Seasonal (2-pasture rotation) grazing, 3-pasture rotation since 1960, regional stocking densities. | perennials | 8.25 | 450 | 45 |
| D | Seasonal (2-pasture rotation) until 1980, 3-pasture rotation until 2010, 8-pasture rotation now. | perennials | 7.2 | 350 | 35 |
| D | Seasonal at regional stocking densities until 2010, 4-pasture rotation since 2010, 0.21 cattle ha$^{-1}$. | perennials | 8.25 | 390 | 35 |
| E | Bull pasture, 0.2 ha$^{-1}$ since 1960 | perennials | 8.25 | 390 | 30 |
| E | Continuous until 1970, seasonal in the period 1970–1990. At regional stocking densities, 4-pasture rotation from 1990 to present at 0.22 cattle ha$^{-1}$. | perennials | 7.8 | 440 | 30 |
| F | Seasonal at regional stocking densities until 1980, 8-pasture rotation in the period 1980–1990,15-pasture rotation at 0.3 cattle ha$^{-1}$ | perennials | 7.8 | 445 | 15 |
| G | Seasonal grazing since 1950, regional stocking densities. | sage | 6.8 | 374 | 40 |
| H | Seasonal grazing 3-pasture rotation since 1950, regional stocking densities. | sage | 7.2 | 440 | 35 |
| I | Continuous until 1970, seasonal in the period 1970–1990 at regional stocking densities, 4-pasture rotation in the period 1990–2000, 16-pasture rotation from 2005 to present at 0.3 animals ha$^{-1}$. | perennials | 8.3 | 425 | 15 |

* Sampling depth (DEPTH) = 20 cm, plant lignin and cellulose content (LIGCELL) = 35.9%, and growing season length (G) = 160 days. ** Regional densities (cattle equivalents/ha): < 1950, 0.245; 1950–1960, 0.204; 1960–1970, 0.186; 1970–1980, 0.124; 1980–1990, 0.122. *** Sage-dominated pastures have considerable carbon in roots and stems of shrubs rather than in grasses and soil organic matter; An APC factor = 0.291, as for annuals, was used.

## 4. Discussion

The results provide one of the first systematic modeling explorations of how different grazing management schemes affect grassland production and soil carbon storage under different climatic conditions. The SNAPGRAZE model includes episodic grazing in the model for forage production and shows its potential sensitivity to periods of stay and rest. The results generally confirm purported benefits of short-duration, high stocking density (SDHSD) schemes over continuous or 2-pasture seasonal grazing systems (Figure 4; Figure 8): production equivalent to or even greater than ungrazed conditions and increased soil carbon. Interestingly, the model predicts that these benefits may be more pronounced at higher stocking densities, in climates with shorter growing seasons, and on forage with lower inherent productive capacity reflected in the maximum community relative growth rate (Figures 4 and 6–8). The call for high stocking densities to accompany increasing pasture numbers and reduced periods of stay is persistent among proponents of SDHSD [55], and the model predictions support that hypothesis.

A major driver of the effects of grazing management on production and soil carbon is growing season length, as driven by climate [56,57]. Longer growing seasons with higher temperatures, at least up to 15 °C *MAT*, and higher rainfall allow greater opportunity for forage to regrow after grazing episodes. Modeled SDHSD schemes provide greater stimulation of production over continuous or seasonal grazing at higher rainfall and temperature in temperate systems but have weaker effects on production at higher rainfall in tropical systems. One reason for the moderated effect on tropical systems with abundant grass cover is that higher $r$ allows strong compensation for grazing even under continuous grazing at relatively high stocking densities. Climate drivers of grazing management effects on soil carbon followed similar patterns (Figure 7) with very little change in SOC expected under SDHSD schemes as compared to continuous grazing except at near maximum stocking densities (Figure 8). These outcomes support recent results that suggest greater benefits for SDHSD grazing in more productive grasslands [56,57].

Another major driver of modeled outcomes is $r$. Relatively little is known about what controls this plant community trait, and it has only recently been measured in field environments [23,25]. This parameter is especially important in determining the rapidity with which plant biomass recovers following a grazing episode and, thus, whether grazing has negative or positive effects on production [23]. The effects of management system on forage production (Figure 4) and soil organic carbon (Figure 7) were more pronounced when $r$ was lower. This pattern may indicate that systems with higher $r$ may be more resilient to continuous grazing, such that rotational grazing schemes offer little advantage in biomass re-growth following grazing. In contrast, systems with low $r$ (and short growing seasons) may depend critically on the use of multi-pasture, short-duration grazing to sustain economically viable livestock densities.

One pervasive result is that SDHSD schemes allow much larger stock densities than continuous or seasonal grazing with either compensation of production relative to ungrazed in colder, drier climates or stimulation of production under climates that are more favorable [17,23,36,56]. This result is striking, because one of the many disagreements about rotational grazing schemes is whether higher stocking densities are viable under SDHSD [14,15,20]. Feasible stocking densities and soil carbon accumulation in SNAPGRAZE were constrained by the assumption that the livestock system must rely on standing forage biomass during the dormant season. This assumption is likely true for pastoralist and other systems that avoid winter supplemental feeding. The SNAPGRAZE model suggests that stocking densities in many cases can be doubled relative to continuous grazing with neutral or even positive stocking density effects on forage production and without declines in forage intake per individual. The feasible ranges predicted by SNAPGRAZE are larger than those typically adopted by producers, especially in cold, dry environments (Figure 4). Likely, producers are conservative and want to avoid the risk of unexpected mid-season forage declines that might require sale of livestock under non-optimal market conditions. SNAPGRAZE could be used to evaluate these scenarios under assumed stochastic parameter values, but that is beyond the scope of this paper [58].

Context dependence may also explain discrepancies in experimental results and expert opinions about the general benefits of rotational grazing. For example, experiments may not detect benefits of SDHSD schemes because they are mostly conducted at moderate densities for continuous grazing conditions [57]. These moderate densities allow the experiments to separate the influence of grazing scheme from the influence of stocking density but avoid densities meaningful to producers and that SDHSD schemes can support but would cause degradation under continuous grazing. Previous reviews generally find no effects of rotational grazing on vegetation cover or standing biomass at these moderate densities, as predicted by SNAPGRAZE. Consequently, the model suggests that controlled experiments have not appropriately tested the capacity of SDHSD schemes as management options.

Overall, the SNAPGRAZE model presented here shows promise as a framework for considering episodic livestock grazing as it might emerge out of different grazing management schemes. Preliminary results match those of a field study in south central Montana, USA, and the outcome of recent reviews that compare the effects of different grazing schemes on vegetation cover and biomass. The analysis of the model does not extend to understanding recovery of degraded rangelands, as I explored only feasible stocking densities that did not reduce initial plant biomass $S_0$, a model parameter that reflects rhizome carbon stores, available leaves for expansion, or other variables that promote the capacity of plants to grow at the onset of the growing season. Recovery from degradation depends on feedbacks between grazing management and shifts in dominant vegetation cover and may involve feedbacks between accumulated soil carbon and $r$ or *ANPPmax*. Such feedbacks are beyond the scope of this model. Nevertheless, SNAPGRAZE provides a framework for such complexities to be incorporated in future research. A final caveat to the model results are that the model lacks known feedbacks between grazing on soil nitrogen availability and leaf tissue nutrients, along with the influence of higher soil organic matter (as reflected by SOC) on potential aboveground production [59,60]. These feedbacks might enhance the positive effects of SDHSD grazing on production and SOC and foster their occurrence at lower stocking densities [20,21] and so might be explored in future research.

**Funding:** The study was supported by US NSF grant DEB DEB1557085.

**Acknowledgments:** The author thanks J.F. Penner for comments.

**Conflicts of Interest:** The author declares no conflict of interest

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
