# Peer review of "Grazing Management, Forage Production and Soil Carbon Dynamics"

_resources, doi:10.3390/resources9040049_

Round 1

Reviewer 1 Report

This study investigated the impacts of grazing on soil carbon dynamic based on the input of plant and herbivory animal residues. I think the topic is interesting and suitable for readers of the journal.

However described methods of soil sampling and analyzing cause some questions.

First, it is not clear how author estimated clay content (CLAY%) in the soil using the standard field ball and soil ribbon method? This field method is qualitative but not quantitative.

The second question concern estimation of soil bulk density. First of all, author measured bulk density only for the depth 0-10cm and write that did not estimate bulk density of the soil on the depth 10-20 cm due to discrete displacement by large gravel on this depth. However he calculate soil carbon stock for this depth based on bulk density for the upper soil layer (0-10cm). It is not correct method because if the gravelty of the soil layer 10-20cm was higher then bulk density of the soil sieved through a 2mm mesh screen (soil that contain organic carbon) will be lower, and as consequence, content of soil organic carbon in this layer will be lower than in the upper soil layer. If it was not possible to take samples on soil bulk density the author should estimate percentage of gravel in that layer and then calculate which part of the volume was filled by soil.

The second remark concern the method of bulk density calculation for the layer 0-10 cm. Author determined the bulk density by dividing soil dry mass by its net volume (Core volume – rock volume). It is not correct if he want calculate the soil carbon density for the territory (tons per hectare). To calculate soil carbon density for the square meter or hectare it is necessary to take bulk density determined by dividing dry mass of soil (sieved through a 2 mm mesh screen) by total core volume (without rock volume removing), because 1 hectare of this soil contain not only soil but the rocks as well. In this case if author remove volume of rock during calculation of bulk density, he receive density of soil between rocks and when he multiply it to the area he receive soil bulk density and then the soil carbon density as it would be in case if soil on this territory had not any rocks. For example: if rocks volume in the soil is 50% of the total volume then calculations like that described by author give 2 times overestimation of carbon density because they will not take into account that 50% of total soil volume on this territory are rocks that do not contain carbon. If rocks take 75% of the volume then overestimation of carbon density will be 4 times. Difference in carbon density with such calculations will depend on rock volume in the soil even if carbon content in the sieved soil will be the similar.

I think that author should improve these calculations to receive real soil carbon density for the studied areas used for the verification of the model.

Author Response

please find the attached reply to your comments.

Reviewer 2 Report

Reviewer comments on resources-743955

The manuscript is fairly well written and reports improvement of a forage production model that includes the positive or negative consequences of grazing management.

All comments and edits are shown at least once on the uploaded pdf, generally using sticky notes. Most critical comments/edits are given below by line.

General:

In many cases, a space is missing between a word and the following in text citation.

Please refer to Allen et al. Grass and Forage Science 66 (1998): 2-28 for correct terminology. Find and replace ‘stock’ with ‘stocking’ when referring to ‘stocking density’.

Make sure abbreviations and acronyms are defined on first use rather than explaining in subsequent text.

Be consistent in defining terms (see lines 82 and 88-89 for an example of different wording for defining SK).

Once a term/parameter abbreviation has been defined, use only that abbreviation throughout the remainder of the manuscript. There is an unnecessary amount of repetition on this point.

Avoid beginning a sentence with an abbreviation, acronym, or numeral. Rephrase sentences if possible; spell out the term or number if necessary.

When a paragraph continues that has an imbedded equation, remove the indentation for the continuation of the paragraph.

Make sure that tables and figures are numbered by chronological citation in the text and that they are placed as near to the first citation as possible. Some are placed multiple pages away and Figure 5 is cited before Figure 4, although they should be renumbered to be appropriate anyway.

Tables and figures should be able to stand alone. Consequently, it is likely not appropriate to refer to tables or figures within tables or figures or to use reference numbers for citations in tables or figures. I have no good advice about how to correct that, but it could be problematic if a reader needs to lift a table or figure for a presentation.

Define all abbreviations used in tables or figures in the table or figure.

For figures containing multiple panes, use the same scale across y-axes and x-axes to enhance differences among the various environments.

Be consistent with format of units, particularly in regard to using ‘/’ or space or dot and superscripts (e.g., line 84, ‘g.g-1.day-1’; line 85, ‘g m-2’; line 121, ‘per m2’).

‘g’ is defined as a biomass term and used as a unit of measure. Does g, when subscripted, always refer to biomass?

Elsewhere t/ha, tons/ha, tons/ha-1 (which is incorrect in itself): be consistent with the units. Ton can be confused as a British or SI unit. I recommend using Mg for metric tonne.

When referring to GRASIM, use the acronym instead of, ‘a model of episodic herbivory’, ‘episodic grazing management model’, etc.

Specific comments:

Line 59 and elsewhere: I’m not sure if this is a correct format for this sort of in text citation. I hope it is and would like clarification by the editors and a statement of such in the instructions to authors or all MDPI journals.

Lines 72-74: This sentence needs to be reworded.

Lines 76-82 and elsewhere: When defining an abbreviation for a term, the usual format is to spell out the term and follow it with the abbreviation in parentheses. This should make the text flow much more smoothly, and be more grammatically correct.

Lines 242-244: Describe the plant survey technique.

Lines 336-337: I’m not certain the statement that annuals dominate many grasslands globally. Citations are necessary to support this statement.

Line 363: Verify the formatting for the sentence: (Equation 10) in Ref[13].

Figure 4: Extend the y-axis to include the intercept for the linear trendline.

Table 4: Format footnotes according to examples in the template.

References:

Reset numbering to begin with 1 and verify all in text citations to be correct in regard to reference number.

Format all references according to the examples of reference types in the template.

Reference #55 is not cited in the text, likely because references begin at #2 in the list.

Author Response

(The authors gave the same response as above.)
